# Bridging Phytochemistry and Cosmetic Science: Molecular Insights into the Cosmeceutical Promise of *Crotalaria juncea* L.

**DOI:** 10.3390/ijms26167716

**Published:** 2025-08-09

**Authors:** Tanatchaporn Aree, Siripat Chaichit, Jintana Junlatat, Kanokwan Kiattisin, Aekkhaluck Intharuksa

**Affiliations:** 1Department of Pharmaceutical Sciences, Faculty of Pharmacy, Chiang Mai University, Chiang Mai 50200, Thailand; tanatchaporn.mp@gmail.com (T.A.); siripat.chaichit@cmu.ac.th (S.C.); 2Faculty of Thai Traditional and Alternative Medicine, Ubon Ratchathani Rajabhat University, Ubon Ratchathani 34000, Thailand; jintana.j@ubru.ac.th

**Keywords:** anti-aging, anti-inflammation, anti-oxidation, anti-tyrosinase, cosmetic, Fabaceae, flavonoids, molecular docking, skin, Sunn hemp

## Abstract

*Crotalaria juncea* L. (Fabaceae: Faboideae), traditionally used as green manure due to its nitrogen-fixing capacity, also exhibits therapeutic potential for conditions such as anemia and psoriasis. However, its cosmetic applications remain largely unexplored. This study examined the phytochemical profiles and biological activities of ethanolic extracts from the root, flower, and leaf of *C. juncea*, focusing on their potential use in cosmetic formulations. Soxhlet extraction with 95% ethanol was employed. Among the extracts, the leaf showed the highest total flavonoid content, while the root contained the highest total phenolic content. The root extract demonstrated the strongest antioxidant activity, as assessed by DPPH, FRAP, and lipid peroxidation assays, along with significant anti-tyrosinase and anti-aging effects via collagenase and elastase inhibition. LC-MS/QTOF analysis identified genistein and kaempferol as the major bioactive constituents in the root extract. Molecular docking confirmed their strong interactions with enzymes associated with skin aging. Additionally, the root extract exhibited notable anti-inflammatory activity. These results suggest that *C. juncea* root extract is a promising multifunctional natural ingredient for cosmetic applications due to its antioxidant, anti-tyrosinase, anti-aging, and anti-inflammatory properties.

## 1. Introduction

Skin aging is an inevitable biological process influenced by both intrinsic and extrinsic factors. Intrinsic factors include hormonal changes, cellular metabolism, genetic predisposition, and other physiological mechanisms, while extrinsic factors encompass ultraviolet (UV) radiation, environmental pollutants, ionizing radiation, chemicals, and toxins [1]. Increasing evidence has highlighted chronic inflammation—termed “inflammaging”—as a major contributor to aging and age-related conditions. Oxidative stress and DNA damage, closely associated with inflammation, are recognized as central drivers of skin aging. To counteract these effects, various strategies such as antioxidant therapy, hormone replacement, and vitamin supplementation have been investigated [2]. Antioxidants, whether produced endogenously or obtained through diet and supplements, function as free radical scavengers, protecting cells from damage caused by reactive oxygen species (ROS) [3]. Concurrently, melanogenesis—the process by which melanocytes produce melanin, the pigment responsible for the coloration of skin, hair, and eyes—involves the enzymatic oxidation of L-tyrosine to L-DOPA and dopaquinone, catalyzed by tyrosinase. Overactivity of tyrosinase can result in hyperpigmentation disorders such as melasma, freckles, and age spots [4,5]. Currently, natural products derived from plants, animals, and microorganisms are extensively explored for their potential anti-aging properties in cosmetic and dermatological applications [6].

Herbal medicine, the oldest form of medical treatment, has played a crucial role in ancient civilizations and continues to be one of the most widely practiced healthcare systems worldwide [7]. Medicinal plants serve diverse functions, including use in food preservation, pharmaceuticals, alternative medicine, and natural therapies. Compared to synthetic compounds, naturally derived substances are generally more biodegradable and environmentally sustainable, contributing to the rising consumer preference for natural antioxidants and nutrients [8]. One notable genus in the Fabaceae family is *Crotalaria*, which has been extensively studied for its phytochemical and pharmacological properties. Numerous *Crotalaria* species exhibit medicinal, antibacterial, antimicrobial, and cytotoxic activities, often attributed to their pyrrolizidine alkaloid content. With approximately 490 species employed in traditional medicine, *Crotalaria* ranks as the second-largest medicinal genus across 46 plant families [9]. In peninsular India, particularly in Karnataka, Andhra Pradesh, Kerala, and Tamil Nadu, around 93 *Crotalaria* species have been documented, many of which are utilized for fiber production, forage, green manure, and medicinal applications [10]. For therapeutic use, *C. albida* roots are traditionally used as a purgative; meanwhile, the whole plant of *C. spectabilis* and the leaves of *C. verrucosa* are applied in the treatment of scabies and impetigo [10].

*Crotalaria juncea* L. (Sunn hemp), a leguminous plant belonging to the Fabaceae family (subfamily Faboideae), is an erect, short-day annual species that grows 1–4 m in height, characterized by a deep taproot system, hairy pods, and bright yellow flowers (Figure 1) [11]. It is extensively cultivated as green manure, forage, and an erosion-control cover crop due to its high nitrogen-fixing capacity [12]. Pharmacological studies have reported its anti-diarrheal effects in castor oil-induced rat models [13] and its antihyperlipidemic activity in Triton-induced hyperlipidemic mice, with significant improvements in lipid profiles at doses of 100 and 200 mg/kg [14]. Despite its broad pharmacological potential, the cosmetic applications of *C. juncea* remain largely unexplored. This study aimed to assess the biological activities of its flower, leaf, and root ethanolic extracts, with particular emphasis on antioxidant, anti-aging, anti-tyrosinase, and anti-inflammatory properties. In addition, phytochemical profiling and molecular docking analyses were performed to identify active constituents and elucidate their mechanisms of action, thereby supporting the potential development of *C. juncea* as a novel bioactive ingredient for cosmetic use.

## 2. Results and Discussion

### 2.1. Appearance and Yield of C. juncea Extracts

The roots, flowers, and leaves of *C. juncea* were extracted using Soxhlet extraction with ethanol. The flower extract yielded a yellow liquid with a distinct aroma, the leaf extracts a green liquid with a characteristic scent, and the root extract a dark brown semi-solid with a unique odor. Among the three plant parts, the leaf extract exhibited the highest extraction yield (16.13%), followed closely by the flower extract (16.10%), while the root extract had a substantially lower yield (3.27%). Ethanol was selected as the extraction solvent due to its broad solubility profile and proven efficacy in isolating a wide range of phytochemicals, making it a commonly used and effective solvent for natural product extraction.

### 2.2. Total Phenolic and Total Flavonoid Contents of C. juncea Extracts

Total phenolic content was assessed using the Folin–Ciocalteu assay, which detects hydroxyl groups in phenolic compounds. These compounds, including flavonoids such as flavanols, flavones, and condensed tannins, are known for their potent antioxidant properties both in vitro and in vivo, primarily through the reduction in oxidative stress. Owing to these bioactivities, phenolic-rich extracts from fruits, herbs, vegetables, cereals, and other plant sources are increasingly applied in the food and nutraceutical industries [12]. The gallic acid standard curve was defined by the equation y = 7.8405x + 0.0319 (R^2^ = 0.9979). Among the extracts, the root exhibited the highest total phenolic content (27.22 ± 0.01 mg gallic acid equivalent [GAE]/g extract), followed by the leaf (23.48 ± 0.3 mg GAE/g extract) and flower (19.10 ± 0.5 mg GAE/g extract). Similarly, flavonoids act as effective scavengers of reactive molecules such as singlet oxygen and reactive oxygen species (ROS). They function by inhibiting ROS formation, chelating trace metals involved in ROS generation, and enhancing endogenous antioxidant defenses. The quercetin standard curve was represented by y = 1.131x − 0.0844 (R^2^ = 0.9932). The leaf extract showed the highest total flavonoid content (43.07 ± 0.02 mg quercetin equivalent [QE]/g extract), followed by the flower (24.20 ± 0.01 mg QE/g extract) and root (23.57 ± 0.01 mg QE/g extract). Polyphenols are typically extracted using polar or semi-polar solvents such as ethanol, methanol, or aqueous mixtures. Their extraction efficiency is influenced by factors including the physicochemical characteristics of the plant matrix, solvent polarity, and extraction duration [15]. While phenolic compounds are predominantly hydrophilic and favor polar solvents, flavonoids vary in polarity, with many being semi-polar or hydrophobic. Thus, careful solvent selection is critical for optimal extraction efficiency.

### 2.3. Antioxidant Activity of C. juncea Extracts

The antioxidant activities of *C. juncea* flower, leaf, and root extracts were assessed using three standard assays: DPPH radical scavenging, ferric reducing antioxidant power (FRAP), and β-carotene bleaching (Table 1). Oxidative stress and radiation exposure are well known for inducing excessive production of free radicals, resulting in cellular and biomolecular damage [16,17]. In the DPPH assay, a lower IC_50_ value indicates stronger radical scavenging capacity. The root extract exhibited the most potent activity (IC_50_ = 0.52 ± 0.06 mg/mL), correlating with its high phenolic content. In the FRAP assay, antioxidant capacity is determined by the reduction of the Fe^3+^-TPTZ complex to Fe^2+^-TPTZ under acidic conditions, producing a blue-colored complex measured at 593 nm. The root extract again demonstrated the highest antioxidant potential (15.21 ± 1.46 mg FeSO_4_/g extract), whereas the flower and leaf extracts showed no significant difference. Lipid peroxidation inhibition was evaluated via the β-carotene bleaching assay, which measures the ability of antioxidants to prevent oxidative degradation of β-carotene by peroxyl radicals generated from linoleic acid oxidation [18,19]. All extracts exhibited strong inhibitory effects comparable to the standard antioxidant Trolox: flower (90.99 ± 1.27%), leaf (89.19 ± 5.84%), root (90.99 ± 7.09%), and Trolox (89.86 ± 6.72%). These results confirm that *C. juncea* extracts possess strong antioxidant activity, primarily attributed to their phenolic and flavonoid constituents. The root extract showed superior performance in both the DPPH and FRAP assays, likely due to its unique phytochemical profile, which may include compounds adapted for protecting subterranean tissues from environmental stressors such as pathogenic fungi and heavy metals [20]. However, the similar efficacy observed across all extracts in the β-carotene bleaching assay suggests a shared capacity for lipid peroxidation inhibition. Phenolic compounds exert antioxidant effects through hydrogen donation to stabilize free radicals and by chelating transition metal ions (e.g., Fe^2+^ and Cu^2+^), thereby inhibiting their pro-oxidant activity. Previous studies have reported that the high levels of tannins and phenolic compounds in *C. juncea* contribute to its notable metal-chelating and radical-scavenging properties [21]. Collectively, these findings highlight the strong antioxidant potential of *C. juncea* root extract, supporting its application as a multifunctional natural agent in cosmetic formulations targeting oxidative stress and skin aging.

### 2.4. Anti-Tyrosinase Activity of C. juncea Extracts

Tyrosinase, a multifunctional oxidase, plays a central role in melanin biosynthesis and is widely distributed across biological systems. Melanin is synthesized by melanocytes located in the stratum basale of the epidermis, primarily in response to ultraviolet (UV) radiation. Upon UV exposure, tyrosinase catalyzes two key reactions: the hydroxylation of L-tyrosine to L-3,4-dihydroxyphenylalanine (L-DOPA), followed by the oxidation of L-DOPA to dopaquinone [22]. Given its pivotal role in melanogenesis, tyrosinase represents a major target in the development of skin-whitening agents, as its inhibition can effectively reduce melanin production and alleviate hyperpigmentation. The tyrosinase inhibitory activity of *C. juncea* extracts was evaluated using a modified dopachrome assay with L-tyrosine and L-DOPA as substrates (Table 2). When L-tyrosine was used, the root and flower extracts exhibited inhibition rates of 33.56 ± 4.21% and 26.62 ± 2.03%, respectively. With L-DOPA as the substrate, the root and leaf extracts showed higher inhibition levels of 44.15 ± 4.03% and 43.91 ± 9.17%, respectively. Kojic acid, used as a positive control, demonstrated strong inhibitory effects in both assays. Previous studies have reported modest tyrosinase inhibition by methanolic (16.12 ± 1.96%) and aqueous (22.45 ± 0.27%) extracts of *C. juncea* shoots [22]. Additionally, three novel compounds isolated from *C. pallida* seeds have shown tyrosinase-inhibitory activity [23]. Flavonoids, which can function as substrates or competitive inhibitors of tyrosinase, have been widely investigated for their depigmenting properties [24]. Various phenolic compounds—including low-molecular-weight phenolics, tannins, and polyphenol derivatives containing multiple aromatic rings—also contribute significantly to tyrosinase inhibition [25]. Collectively, these findings suggest that the root extract of *C. juncea* exhibits moderate tyrosinase inhibitory activity, likely attributed to its phenolic and flavonoid content. This supports its potential utility as a natural agent in cosmetic formulations designed to regulate skin tone and reduce hyperpigmentation.

### 2.5. Anti-Aging Activity of C. juncea Extracts

The anti-aging potential of *C. juncea* extracts was evaluated by assessing their inhibitory effects on three key enzymes involved in the degradation of the skin’s extracellular matrix (ECM): collagenase, elastase, and hyaluronidase. Collagen, the primary structural protein in the dermis, is degraded by collagenase, leading to fragmentation and wrinkle formation [26]. Elastase targets elastin, a protein essential for maintaining skin elasticity, while hyaluronidase breaks down hyaluronic acid—a critical hydrating polysaccharide—thereby increasing tissue permeability and accelerating skin aging [27]. The anti-aging activities of the extracts are summarized in Table 3. The root extract exhibited the strongest collagenase inhibition (91.60 ± 0.66%), comparable to the positive control, ascorbic acid, and significantly higher than the flower and leaf extracts (*p* < 0.05). For elastase inhibition, the root and leaf extracts demonstrated moderate activity (67.17 ± 9.34% and 65.04 ± 13.61%, respectively), though both were lower than ascorbic acid (86.51 ± 11.70%). In terms of hyaluronidase inhibition, the root extract again showed the highest activity (8.50 ± 2.53%), although the differences among the extracts were not statistically significant (*p* > 0.05). These findings underscore the strong anti-aging potential of the *C. juncea* root extract, particularly through its pronounced inhibition of collagenase, suggesting its potential utility as a natural anti-aging ingredient in cosmetic formulations.

### 2.6. In Vitro Anti-Inflammatory Activity of C. juncea Extracts

Lipoxygenases (LOXs) are a family of enzymes that catalyze the oxygenation of polyunsaturated fatty acids containing a cis-1,4-pentadiene structure, leading to the formation of hydroperoxides. These hydroperoxides are subsequently converted into pro-inflammatory lipid mediators, including leukotrienes and hydroxy-eicosatetraenoic acids (HETEs), which play crucial roles in inflammatory processes, particularly those affecting the skin. LOX-mediated pathways are implicated in skin inflammation, oxidative stress, and photoaging. Chronic inflammation and elevated levels of reactive oxygen species (ROS) contribute to skin aging by promoting collagen degradation, impairing the skin barrier, and activating matrix metalloproteinases (MMPs). Therefore, inhibition of LOX activity is considered a promising strategy for mitigating inflammation and oxidative damage, with relevance to anti-aging and anti-inflammatory cosmetic applications [28]. The LOX inhibitory activity of *C. juncea* extracts is illustrated in Figure 2. Both the leaf and root extracts exhibited significant inhibitory effects (57.32 ± 11.14% and 56.60 ± 9.77%, respectively), significantly greater than that of the flower extract (*p* < 0.05). Indomethacin, used as a positive control, demonstrated the highest inhibition (76.31 ± 6.71%). Similar anti-LOX activity has been reported in related species such as *C. longipes*, further supporting the potential of *C. juncea* as a source of anti-inflammatory agents [29]. The observed activity is likely attributable to the presence of bioactive phytochemicals such as flavonoids and alkaloids. In healthy skin, a balance between proteolytic enzymes (e.g., serine proteases) and their natural inhibitors (e.g., α_1_-antitrypsin, secretory leukocyte protease inhibitor [SLPI]) are essential for maintaining structural integrity. Disruption of this balance—particularly excessive protease activity—can lead to degradation of critical proteins like collagen and elastin, resulting in inflammation and tissue damage [28]. The proteinase inhibitory activities of *C. juncea* extracts are also presented in Figure 2. The flower, leaf, and root extracts exhibited comparable inhibitory effects (67.82 ± 3.29%, 70.91 ± 1.33%, and 71.16 ± 1.49%, respectively), with no statistically significant differences among them (*p* > 0.05). Indomethacin again showed the highest inhibition (94.97 ± 2.47%). The anti-inflammatory and anti-proteolytic activities of *C. juncea* are likely linked to its phytochemical composition, particularly flavonoids, which are known to modulate inflammatory signaling pathways. The bioactivity observed in the leaf extract aligns with previous studies on green leafy vegetables rich in polyphenols, flavonoids, and carotenoids, which have demonstrated potent anti-inflammatory properties [30]. Collectively, these findings indicate that *C. juncea*, especially its root and leaf extracts, possesses dual anti-inflammatory activity through the inhibition of both LOX and protease enzymes. This highlights its potential for inclusion in cosmetic formulations aimed at addressing skin inflammation, irritation, and aging.

### 2.7. Anti-Inflammatory Activity of C. juncea Extract in Murine Macrophage

A comprehensive evaluation of *C. juncea* root extract was undertaken due to its consistently superior performance in antioxidant, anti-aging, tyrosinase inhibition, and anti-inflammatory assays. Consequently, the root extract was selected for further investigation, focusing on its cytotoxicity, nitric oxide inhibition, and modulation of inflammation-related gene expression in murine macrophages.

#### 2.7.1. Cytotoxicity Assessment

The cytotoxicity of *C. juncea* root extract was evaluated in RAW 264.7 macrophage cells. The maximum non-cytotoxic concentration (MNTC) was defined as the concentration at which cell viability remained ≥ 80% [31]. The extract exhibited a dose-dependent effect on cell viability, with concentrations up to 300 µg/mL maintaining cell viability above the threshold. However, at 600 µg/mL, viability declined below 80%. Similar findings have been reported in previous studies, where *C. juncea* flower extracts showed no cytotoxicity in L929 cells at concentrations ranging from 1 to 100 µg/mL, regardless of the ethanol concentration used for extraction [32]. Based on these results, concentrations below 300 µg/mL were selected for subsequent anti-inflammatory assays.

#### 2.7.2. Nitric Oxide Inhibition

Nitric oxide (NO) is a key pro-inflammatory mediator predominantly produced by activated macrophages [33]. To evaluate the NO inhibitory activity of *C. juncea* root extract, RAW 264.7 cells were stimulated with lipopolysaccharide (LPS). The extract significantly reduced NO production in a dose-dependent manner, with 100 µg/mL achieving 20.58 ± 1.93% inhibition (Figure 3). Although the inhibitory effect was lower than that of aminoguanidine, the positive control, the results support the extract’s potential to modulate macrophage-mediated inflammatory responses. These findings are consistent with previous studies reporting dose-dependent NO inhibition by petroleum ether extracts of *C. juncea* seeds, along with in vivo anti-inflammatory activity in rat models [34].

#### 2.7.3. Anti-Inflammatory Activity of *C. juncea* Extract

Inflammatory cytokines such as IL-1β, IL-6, and TNF-α play pivotal roles in mediating immune responses and tissue inflammation [35,36,37]. In this study, LPS-stimulated RAW 264.7 macrophages exhibited significantly elevated levels of these cytokines, which were markedly suppressed following treatment with *C. juncea* root extract at 100 µg/mL (Figure 4). The extract effectively downregulated the expression of IL-1β, IL-6, and TNF-α, although the extent of inhibition was lower than that observed with indomethacin (50 µg/mL), the positive control. These findings align with previous reports demonstrating that isoflavones such as genistein and daidzein inhibit NF-κB activation, nitric oxide production, and pro-inflammatory cytokine expression [38]. The presence of genistein in the root extract suggests it may contribute significantly to the observed anti-inflammatory effects.

These findings further support the anti-inflammatory potential of *C. juncea* root extract, demonstrated through the inhibition of nitric oxide production and downregulation of key pro-inflammatory cytokines. When considered alongside its previously established antioxidant and anti-aging activities, the root extract of *C. juncea* presents itself as a promising multifunctional candidate for cosmetic and therapeutic applications aimed at combating inflammation-induced skin aging.

### 2.8. Characterization of the Phytochemical Composition of C. juncea Extract Using Liquid Chromatography–Mass Spectrometry–Quadrupole Time-of-Flight (LC-MS/QTOF)

The phytochemical composition of *C*. *juncea* root extract was analyzed using liquid chromatography–mass spectrometry with quadrupole time-of-flight detection (LC-MS/QTOF). The analysis revealed a diverse range of phytochemical signals, with several peaks tentatively assigned to flavonoids and their derivatives based on accurate mass, isotopic pattern, and database matching. Selected putative constituents are listed in Table 4. This study focused primarily on flavonoids due to their well-established relevance in cosmetic applications. Among the tentatively identified flavonoids were genistein (RT = 7.569), kaempferol (RT = 6.299), dihydroquercetin (RT = 4.249), kaempferol-3-O-glucoside (RT = 4.102), and chrysin (RT = 6.540) as shown in Figure 5 and Figure 6. In addition to flavonoids, the analysis also suggested the presence of sugars, carbohydrates, and fatty acids, which may contribute to the overall bioactivity and formulation potential of the extracts.

The high abundance of flavonoid-related peaks supports their contribution to the extract’s antioxidant activity. Flavonoids represent a major class of phenolic compounds known for their antioxidant, anti-inflammatory, and anti-aging activities. Isoflavones, such as genistein, are a subgroup of flavonoids classified as phytoestrogens and are predominantly synthesized by plants in the Fabaceae family [39]. These findings suggest that *C. juncea* root extract may serve as a promising source of bioactive constituents for cosmeceutical applications, although further studies using reference standards and structural elucidation techniques are required to confirm the compound identities.

### 2.9. Molecular Docking of Genistein and Kaempferol Against Anti-Aging Enzymes

Molecular docking analysis of the major bioactive compounds identified in *C. juncea* root extract—specifically genistein and kaempferol—was conducted against three key skin aging-related enzymes: collagenase, elastase, and hyaluronidase. The results revealed notable differences in binding energies and interaction profiles, providing insights into their potential inhibitory mechanisms (Figure 7a). Both genistein and kaempferol exhibited favorable binding affinities toward all three enzymes, suggesting strong molecular interactions. Genistein showed the highest binding affinity for elastase (−10.1 kcal/mol), followed by hyaluronidase (−7.9 kcal/mol) and collagenase (−7.5 kcal/mol). In contrast, kaempferol demonstrated consistently strong and balanced interactions, with binding energies of −9.4 kcal/mol for elastase and −8.2 kcal/mol for both hyaluronidase and collagenase. Although genistein displayed the most favorable binding to elastase, kaempferol’s uniform affinity across all targets suggests a broader inhibitory potential. These results support the role of both compounds—particularly kaempferol—as promising inhibitors of enzymes involved in extracellular matrix degradation, further reinforcing the anti-aging potential of *C. juncea* root extract in cosmetic applications.

Interaction pattern analysis of genistein and kaempferol with skin aging-related enzymes revealed distinct molecular binding behaviors that further support their bioactivity profiles. For collagenase, genistein formed hydrogen bonds with residues R214 and E219 and engaged in hydrophobic interactions with V215, R214, and S239. In contrast, kaempferol demonstrated stronger binding by coordinating with the catalytic zinc ion and forming hydrogen bonds with H218, suggesting a more specific and stable interaction with the enzyme’s active site (Figure 7b). In the elastase docking model, genistein interacted with V136 and F138 through hydrogen bonding and hydrophobic contacts. Kaempferol, however, formed a broader interaction network involving L82 and A83, including multiple hydrogen bonds, which may enhance stabilization within the enzyme’s binding pocket (Figure 7c). For hyaluronidase, both compounds interacted with E131 and D206. Kaempferol exhibited additional hydrogen bonds with R265 and S245, resulting in a more extensive interaction network and stronger binding affinity (Figure 7d). Previous studies have linked genistein’s antioxidant and anti-collagenase activities to its structural features—specifically, the presence of hydroxyl groups at the C-3 and C-5 positions, an ortho-dihydroxy configuration in the B ring, and a 2,3 double bond conjugated with a 4-oxo function in the C ring [40,41]. Genistein has also been shown to downregulate matrix metalloproteinase-1 (MMP-1), a key enzyme involved in collagen degradation, and to stimulate collagen synthesis in dermal cells via activation of the TGF-β1 signaling pathway [42]. Moreover, it has been reported to inhibit UV-induced MMP expression [43] and significantly reduce reactive oxygen species (ROS) levels in fibroblasts [44]. Similarly, kaempferol has been shown to inhibit MMP-1 and MMP-3, the key enzymes responsible for collagen breakdown [45]. It also effectively counteracts oxidative stress and prolongs fibroblast lifespan [46]. Furthermore, kaempferol has been reported to upregulate SIRT1, a longevity-associated protein involved in delaying cellular senescence [47]. Its inhibitory effects on collagenase, elastase, and hyaluronidase are attributed to the hydroxyl group at the C-3 position, which facilitates effective hydrogen bonding with the active sites of these enzymes [48].

In summary, while genistein exhibited selective and potent binding—particularly to elastase—kaempferol demonstrated more balanced and stronger binding across all three enzymes. Its complex interaction networks, including coordination with catalytic residues, underscore its potential as a multi-target inhibitor of extracellular matrix-degrading enzymes. These findings highlight kaempferol as a promising skin-protective and anti-aging agent suitable for cosmeceutical applications.

Phytochemical analysis of *C. juncea* extracts from the flowers, leaves, and roots revealed that the root extract exhibited the most significant biological activities, including strong antioxidant, anti-tyrosinase, anti-aging, and anti-inflammatory effects. These bioactivities are consistent with phytochemical profiling results, which identified genistein and kaempferol as major flavonoids in the root extract. Notably, genistein—an isoflavone well recognized for its dermatological benefits—has been shown to enhance skin elasticity, reduce wrinkle formation, and protect against photoaging. These effects are mediated through multiple pathways, including upregulation of vascular endothelial growth factor (VEGF) and transforming growth factor-beta (TGF-β), stimulation of collagen synthesis, and inhibition of collagen degradation via increased tissue inhibitor of metalloproteinases (TIMP) expression and decreased matrix metalloproteinase (MMP) activity [49,50]. Supporting these findings, molecular docking simulations demonstrated strong binding affinities of both genistein and kaempferol to key enzymes involved in skin aging and inflammatory signaling. Collectively, these results highlight the potential of *C. juncea* root extract as a rich source of bioactive compounds for the development of multifunctional cosmetic formulations with anti-aging and anti-inflammatory properties.

## 3. Materials and Methods

### 3.1. Reagents

2,2-diphenyl-1-picrylhydrazyl radical (DPPH), 2,4,6-Tris(2-pyridyl)-s-triazine (TPTZ), clostridium histolyticum collagenase (E.C.3.4.23.3), porcine pancreatic elastase (PE–E.C.3.4.21.36), bovine testis hyaluronidase (E.C.3.2.1.3.5), synthetic peptide of N-[3-(2-furyl) acryloyl]-Leu-Gly-Pro–Ala (FALGPA), N-succinyl-Ala–Ala–Ala–p-nitroanilide (AAAPVN), hyaluronic acid, tyrosinase from mushroom, linoleic acid, perchloric acid, Trolox, kojic acid, and bovine serum albumin (BSA) were purchased from Sigma Aldrich (St. Louis, MO, USA). Acetic acid (CH_3_COOH), di-sodium hydrogen phosphate (Na_2_PO_4_), ethanol, methanol, dimethyl sulfoxide (DMSO), hydrochloric acid (HCl), and sodium dihydrogen phosphate dihydrate (NaH_2_PO_4_·2H_2_O) were purchased from Labscan Asia Co., Ltd. (Bangkok, Thailand). Xylenol orange was purchased from QReC™ (Auckland, New Zealand). Polysorbate 80 was purchased from Namsiang Co., Ltd. (Bangkok, Thailand). Potassium persulfate (K_2_S_2_O_8_), sodium acetate trihydrate (CH_3_COONa·3H_2_O), and ferrous sulfate (FeSO_4_) was purchased from Loba Chemie Pvt Ltd. (Mumbai, India). Folin–Ciocalteu reagent and ferric chloride (FeCl_3_) were purchased from Merck (Darmstadt, Germany). Ammonium thiocyanate [NH_4_SCN] was purchased from KEMAUS (Cherrybrook, Australia).

### 3.2. Preparation of C. juncea Extracts

The roots, flowers, and leaves of *C. juncea* were cultivated and collected from the Faculty of Pharmacy, Chiang Mai University. The plant materials were authenticated based on morphological characteristics by Ms. Wannaree Charoensup, a botanist at the same institution. A voucher specimen was deposited in the Herbarium of the Faculty of Pharmacy, Chiang Mai University (Herbarium code: CMU; specimen number: 0023395). The collected plant materials were dried in a hot air oven at 50 °C (Universal Oven Memmert UN 55, Schwabach, Germany) until fully dehydrated. The dried samples were then ground into a fine powder using an electric blender. Each plant part was subjected to Soxhlet extraction with 95% *v*/*v* ethanol for 20 h. The resulting extracts were concentrated by solvent removal using a rotary evaporator (Buchi R-300, St. Gallen, Switzerland). The percentage yield of each extract was calculated using the following equation:Yield (%) = [Weight of dry extract (g)/Weight of dry powdered material (g)] × 100

### 3.3. Chemical Compounds Analysis

#### 3.3.1. Determination of Total Phenolic Content

The total phenolic content of each extract was determined using the Folin–Ciocalteu colorimetric method [51], with gallic acid serving as the standard. Each extract was prepared at a concentration of 1 mg/mL in deionized (DI) water. A 0.5 mL aliquot of each sample was mixed thoroughly with 2 mL of 10% *v*/*v* Folin–Ciocalteu reagent. Subsequently, 4 mL of 7.5% *w*/*v* sodium carbonate solution was added to the mixture, which was then incubated at room temperature for 30 min. The absorbance was measured at 765 nm using a UV-Vis spectrophotometer (UV-2450, Shimadzu, Duisburg, Germany). The total phenolic content was expressed as milligrams of gallic acid equivalent (mg GAE) per gram of extract.

#### 3.3.2. Determination of Total Flavonoid Content

The total flavonoid content of each extract was determined using the aluminum chloride colorimetric method [51]. Each extract was prepared at a concentration of 10 mg/mL in deionized (DI) water. A 1 mL aliquot of each sample was transferred into a 10 mL volumetric flask containing 4 mL of distilled water. Subsequently, 0.3 mL of 5% *v*/*v* sodium nitrite solution was added. After 5 min, 0.3 mL of 10% *w*/*v* aluminum chloride solution was added, and the mixture was allowed to stand for 1 min. Then, 2 mL of 1 M sodium hydroxide and 2.4 mL of distilled water were added. The absorbance was measured at 510 nm using a UV-Vis spectrophotometer (UV-2450, Shimadzu, Duisburg, Germany). The total flavonoid content was expressed as milligrams of quercetin equivalent (mg QE) per gram of extract.

### 3.4. Antioxidant Activity Tests

#### 3.4.1. DPPH Radical Scavenging Assay

The DPPH radical scavenging activity of each extract was assessed using the standard DPPH assay [51]. Extracts were prepared at various concentrations in 95% *v*/*v* ethanol. In a 96-well microplate, 20 µL of each extract solution was mixed with 180 µL of DPPH solution. The mixtures were incubated in the dark at room temperature for 30 min. Absorbance was measured at 520 nm using a microplate reader (BMG Labtech, SPECTROstar Nano, Ortenberg, Germany). Trolox was used as the positive control. The percentage of radical scavenging activity was calculated using the following equation:% inhibition = [(Control − Blank of control) − (Sample − Blank of sample)/(Control − Blank of control)] × 100
where Control is the absorbance of the DPPH solution, Blank of control is the absorbance of the solvent, Sample is the absorbance of the sample with DPPH, and Blank of sample is the absorbance of the sample without DPPH. The results are expressed as the half-maximal inhibitory concentration (IC_50_) value, representing the concentration of extract required to scavenge 50% of DPPH radicals.

#### 3.4.2. Ferric Reducing Antioxidant Power (FRAP) Assay

The ferric reducing antioxidant power (FRAP) assay was performed to evaluate the reducing capacity of each extract, following the method described by Nitthikan et al. (2018) [52]. The FRAP reagent was freshly prepared by combining 50 mL of 300 mM acetate buffer, 5 mL of 10 mM TPTZ (2,4,6-tripyridyl-s-triazine) dissolved in 40 mM hydrochloric acid (37% *v*/*v*), and 5 mL of 20 mM ferric chloride solution. For the assay, 20 µL of each extract (prepared at 1 mg/mL) was mixed with 180 µL of FRAP reagent in a 96-well microplate. The mixture was incubated at 37 °C for 4 min, after which the absorbance was measured at 593 nm using a microplate reader (BMG Labtech, SPECTROstar Nano, Ortenberg, Germany). Trolox was used as the positive control. The FRAP value was determined using a standard curve generated with ferrous sulfate and expressed as milligrams of ferrous sulfate equivalents (mg FeSO_4_/g extract).

#### 3.4.3. β-Carotene Bleaching Assay

The β-carotene bleaching inhibitory activity of each extract was evaluated using the method described by Barros et al. (2007) [53]. A β-carotene–linoleate emulsion was prepared by dissolving β-carotene (5 mg/mL) in chloroform, followed by the addition of 185 µL of Tween 20 and 25 µL of linoleic acid. The chloroform was then removed either under vacuum or by allowing it to evaporate completely at room temperature. Once the solvent was fully evaporated, 50 mL of distilled water was added, and the mixture was vigorously shaken to form a stable emulsion. In a 96-well microplate, 180 µL of the prepared emulsion was mixed with 20 µL of each extract (1 mg/mL). The absorbance was measured at 450 nm using a microplate reader (BMG Labtech, SPECTROstar Nano, Ortenberg, Germany). Absorbance readings were recorded immediately at time zero and subsequently at 30 min intervals for up to 120 min. The β-carotene bleaching inhibitory activity was calculated using the following equation:% inhibition = [(Control − Blank of control) − (Sample − Blank of sample)/(Control − Blank of control)] × 100
where Control is the absorbance of the β-carotene emulsion, Blank of control is the absorbance of the solvent, Sample is the absorbance of the test extract with the β-carotene emulsion, and Blank of sample is the absorbance of the extract without the β-carotene emulsion. This method evaluates the ability of extracts to inhibit lipid peroxidation by preserving β-carotene levels in the emulsion.

#### 3.4.4. Anti-Tyrosinase Activity

Tyrosinase inhibitory activity was evaluated using the dopachrome assay, as described by Kiattisin et al. (2016) [54], employing both L-tyrosine and L-DOPA as substrates. Each extract was prepared at a concentration of 2.5 mg/mL in 20% *v*/*v* Tween 20. In a 96-well microplate, 70 µL of each extract solution was mixed with 70 µL of phosphate buffer (pH 6.5) and 70 µL of mushroom tyrosinase solution (50 units/mL). The mixture was incubated at room temperature for 10 min. Subsequently, 70 µL of 2.5 mM L-tyrosine or L-DOPA in phosphate buffer was added to each well, followed by another incubation at room temperature for 20 min. Absorbance was measured at 450 nm using a microplate reader (BMG Labtech, SPECTROstar Nano, Ortenberg, Germany). Kojic acid was used as the positive control. Tyrosinase inhibition was calculated using the following equation:Inhibition (%) = [(Absorbance control − Absorbance sample)/Absorbance control] × 100
where Absorbance control refers to the absorbance of the reaction mixture containing phosphate buffer, tyrosinase enzyme, and substrate (without extract), and Absorbance sample refers to the absorbance of the reaction containing the test extract, enzyme, and substrate. This assay quantifies the ability of extracts to inhibit tyrosinase activity, which is relevant for applications in hyperpigmentation and skin-whitening products.

### 3.5. Determination of Anti-Aging Activity

#### 3.5.1. Collagenase Inhibitory Activity

Collagenase inhibitory activity of each extract was evaluated following the method described by Phumat et al. (2023) [55]. The collagenase enzyme solution was prepared in 50 mM tricine buffer (pH 7.5) containing 400 mM sodium chloride and 10 mM calcium chloride. The substrate used was 2 mM N-[3-(2-furyl)acryloyl]-Leu-Gly-Pro-Ala (FALGPA), a synthetic peptide commonly employed in collagenase assays. Each extract was prepared at a concentration of 0.5 mg/mL in 20% *v*/*v* DMSO. The extract solution was added to the collagenase enzyme solution and incubated at room temperature for 15 min. The reaction was initiated by adding the FALGPA substrate, and the absorbance was measured at 340 nm in kinetic mode using a microplate reader (BMG Labtech, SPECTROstar Nano, Ortenberg, Germany). Ascorbic acid was used as the positive control. The percentage of collagenase inhibition was calculated using the following equation:Collagenase inhibition (%) = [(Slope control − Slope sample)/Slope control] × 100
where Slope control refers to the rate of absorbance change in the control reaction containing tricine buffer, collagenase, and substrate, and Slope sample refers to the rate of absorbance change in the reaction containing the extract, collagenase, and substrate. This assay reflects the ability of test extracts to inhibit collagen degradation, a key factor in skin aging.

#### 3.5.2. Elastase Inhibitory Activity

Elastase inhibitory activity of each extract was evaluated following the method described by Theansungnoen et al. (2022) [27]. The assay measured the formation of a chromogenic product resulting from the enzymatic hydrolysis of the synthetic substrate N-succinyl-Ala-Ala-Ala-p-nitroanilide (AAAVPN). Each extract was prepared at a concentration of 1 mg/mL in 20% *v*/*v* dimethyl sulfoxide (DMSO). The assay mixture contained 100 mM Tris-HCl buffer (pH 8.0), elastase enzyme, and the test extract. The mixture was incubated at room temperature for 20 min, followed by the addition of 4.4 mM AAAVPN substrate. After a second incubation for 20 min, the absorbance was recorded at 410 nm in kinetic mode using a microplate reader (BMG Labtech, SPECTROstar Nano, Ortenberg, Germany). Ascorbic acid was used as the positive control. Elastase inhibitory activity was calculated using the equation:Elastase inhibition (%) = [(Slope control − Slope sample)/Slope control] × 100
where Slope control is the rate of absorbance change in the control reaction (without extract), and Slope sample is the rate in the presence of the extract.

#### 3.5.3. Hyaluronidase Inhibitory Activity

Hyaluronidase inhibitory activity was determined using the turbidimetric method reported by Phumat et al. (2023) [55]. Tannic acid was used as a positive control. Hyaluronidase enzyme (2.0 mg/mL) was diluted in an enzymatic diluent composed of PBS (pH 7.0), 77 mM sodium chloride, and 0.01% BSA, and prepared at 25 °C. The substrate solution (0.03% hyaluronic acid in PBS, pH of 5.3) was incubated at 80 °C to ensure complete dissolution. Each extract was prepared at 0.5 mg/mL in 20% *v*/*v* DMSO. For the assay, 50 µL of extract was mixed with 100 µL of diluted hyaluronidase enzyme and incubated at 37.5 °C for 10 min. Then, 100 µL of 0.03% hyaluronic acid solution was added, followed by a further incubation at 37.5 °C for 45 min. Finally, 1 mL of acetic albumin solution (pH of 3.75; containing 24 mM sodium acetate, 79 mM acetic acid, and 0.1% BSA) was added and mixed at 25 °C. The turbidity was measured at 600 nm using a microplate reader (BMG Labtech, SPECTROstar Nano, Ortenberg, Germany). Hyaluronidase inhibition was calculated using the following equation:Hyaluronidase inhibition (%) = [Absorbance control/Absorbance sample] × 100
where Absorbance control is the absorbance of the reaction mixture without the extract, and Absorbance sample is the absorbance in the presence of the test extract. This assay evaluates the ability of extracts to inhibit the degradation of hyaluronic acid, a key molecule in skin hydration and structural integrity.

### 3.6. In Vitro Anti-Inflammatory Activity Tests

#### 3.6.1. Lipoxygenase Inhibitory Activity

The lipoxygenase (LOX) inhibitory activity of each extract was evaluated using a modified method based on Preedalikit et al. [56]. Extracts and the positive control, indomethacin, were dissolved in methanol. The LOX enzyme was dissolved in Tris-HCl buffer (pH 7.4). In a 96-well microplate, 20 µL of extract solution (1 mg/mL) was mixed with 20 µL of enzyme solution and 30 µL of linoleic acid. Subsequently, FOX reagent (0.20 mM xylenol orange and 2 mM ferrous sulfate in 110 mM perchloric acid and methanol) was added. The absorbance was measured at 560 nm using a microplate reader (BMG Labtech, SPECTROstar Nano, Ortenberg, Germany). LOX inhibition (%) was calculated using the following equation:% inhibition = [(Control − Sample)/Control] × 100
where Control is the absorbance of the mixture with enzyme, linoleic acid, and FOX reagent, and Sample is the absorbance of the reaction containing the extract.

#### 3.6.2. Proteinase Inhibitory Activity

Proteinase inhibition was assessed following a modified protocol from Preedalikit et al. [56]. Diclofenac sodium was used as the positive control. The reaction mixture included trypsin, Tris-HCl buffer (pH of 7.4), and extract solution (1 mg/mL). After a 5 min incubation at 37 °C, 0.3% *w*/*v* bovine serum albumin (BSA) was added, and the mixture was further incubated for 20 min. The reaction was terminated with 5% *w*/*v* perchloric acid. After centrifugation, the absorbance of the supernatant was measured at 660 nm using a microplate reader. Proteinase inhibition (%) was calculated as follows:Proteinase inhibition (%) = [(Control − Sample)/Control] × 100
where Control is the absorbance of the full reaction mixture without extract, and Sample is the absorbance in the presence of the extract.

### 3.7. Anti-Inflammatory Activity in Murine Macrophage

#### 3.7.1. Cell Culture

RAW 264.7 murine macrophages were obtained from PromoCell (Heidelberg, Germany) and cultured in Dulbecco’s Modified Eagle Medium (DMEM; Gibco, Waltham, MA, USA) supplemented with 10% heat-inactivated calf serum (HyClone, Logan, UT, USA) and 1% penicillin-streptomycin (100 U/mL–100 μg/mL; Gibco, USA). Cells were maintained at 37 °C in a 5% CO_2_ humidified incubator (Thermo Scientific, Waltham, MA, USA).

#### 3.7.2. Cytotoxicity Assay

The cytotoxic effects of *C. juncea* root extract were evaluated using the MTT assay [57]. RAW 264.7 cells were treated with various concentrations (300, 600, 1200, 2400, and 4800 µg/mL) of the extract and incubated for 24 h. Cell viability was determined based on mitochondrial dehydrogenase reduction in MTT to formazan. Absorbance was measured at 570 nm. Cell viability was calculated as follows:Cell viability (%) = [A/B] × 100%
where A is the absorbance of treated cells and B is the absorbance of control cells (treated with incomplete medium).

#### 3.7.3. Nitric Oxide (NO) Inhibition Assay

RAW 264.7 cells (1.5 × 10^5^ cells/mL) were pre-treated with lipopolysaccharide (LPS, 50 ng/mL) for 24 h. NO production was quantified by measuring nitrite levels in the culture medium using Griess reagent, comprising 1% sulfanilamide and 0.1% naphthyl ethylenediamine dihydrochloride in 2.5% phosphoric acid. A 100 µL aliquot of medium was mixed with 100 µL of Griess reagent and incubated for 10 min at room temperature. Absorbance was measured at 540 nm using a microplate reader (EZ Read 2000, Cambridge, UK). Aminoguanidine was used as a positive control. The percentage of NO inhibition was calculated as follows:Inhibition (%) = [(Absorbance Control − Absorbance Sample)/Absorbance Control] × 100

#### 3.7.4. Inflammatory-Related Gene Expression

RAW 264.7 cells were seeded in 12-well plates and treated with various concentrations of *C. juncea* root extract or the positive control. After a 24 h incubation at 37 °C in a 5% CO_2_ atmosphere, cells were stimulated with LPS and incubated for an additional 24 h. Total RNA was extracted using the PureLink RNA Mini Kit (Invitrogen, Waltham, MA, USA). First-strand cDNA synthesis was performed using the iScript cDNA Synthesis Kit (Bio-Rad, Hercules, CA, USA). PCR amplification was carried out using MyTaq Red Mix (Bioline, London, UK) and specific primers (Macrogen, Seoul, Republic of Korea). PCR was conducted in a thermal cycler (Heal Force, Shanghai, China) with 30 cycles. PCR products were separated by 1.5% agarose gel electrophoresis and visualized using ViSafe Green gel stain (Vivantis, Kuala Lumpur, Malaysia). Gel band intensities were analyzed using a gel documentation system (BioStep, Luckenwalde, Germany). Expression levels of IL-6 and TNF-α mRNA were quantified and normalized against β-actin.

### 3.8. Characterization of Phytochemical Composition of C. juncea Extracts Using Liquid Chromatography–Mass Spectrometry–Quadrupole Time-of-Flight (LC-MS/QTOF)

For sample preparation, *C. juncea* root extract was prepared at a concentration of 10 mg/mL in 95% ethanol, with 25 ng/mL sulfadimethoxine included as an internal standard. The mixture was centrifuged at 14,000 rpm for 10 min, and the resulting supernatant was transferred into LC-MS vial for analysis. Liquid chromatography (LC) separation was performed at a flow rate of 0.4 mL/min using a time-based gradient elution program. Mobile phase A consisted of 0.1% (*v*/*v*) formic acid in water, while mobile phase B consisted of 0.1% (*v*/*v*) formic acid in acetonitrile. The gradient was applied as follows: t = 0 min, 0% B; t = 0.5 min, 0% B; t = 10.5 min, 55% B; t = 12.5 min, 75% B; t = 14 min, 100% B; t = 17 min, 100% B. The system was re-equilibrated with 0% B from t = 17.5 min to t = 20.5 min, with a total run time of 23 min. Mass spectrometry (MS) data were acquired using an Agilent LC-QTOF 6545XT Mass Spectrometer (Agilent Technologies, Santa Clara, CA, USA), equipped with a Dual AJS ESI ion source. Instrument parameters were set as follows: drying gas temperature at 325 °C, drying gas flow rate at 13 L/min, nebulizer pressure at 45 psi, sheath gas temperature at 275 °C, and sheath gas flow rate at 12 L/min. The capillary voltage (VCap) was set to 4000 V, and the nozzle voltage to 3000 V for both positive and negative ionization modes. Internal reference masses were continuously infused using an Agilent 1260 isocratic pump (Santa Clara, CA, USA). Reference mass ions were *m*/*z* 112.98558700 and *m*/*z* 1033.98810900 for negative mode, and *m*/*z* 121.05087300 and *m*/*z* 922.00979800 for positive mode, ensuring accurate mass calibration throughout the analysis.

### 3.9. Molecular Docking

Molecular docking simulations were carried out to investigate the interactions between bioactive compounds from *Crotalaria juncea* root extract—specifically genistein and kaempferol—and key skin aging-related enzymes. The molecular structures of the ligands were geometrically optimized using Gaussian16 [58]. Docking studies were performed using AutoDock Vina (version 1.2.0) [59], simulating the binding of these marker compounds to target enzymes. Protein structures for collagenase (PDB ID: 1CGL), elastase (PDB ID: 1JIZ), and hyaluronidase (PDB ID: 2PE4) were retrieved from the RCSB Protein Data Bank. Prior to docking, all co-crystallized ligands and water molecules were removed, with the exception of essential zinc and calcium ions. Hydrogen atoms were added, and partial atomic charges were assigned to the protein structures. Docking grids were defined according to the coordinates of the native ligand-binding sites. Validation of the docking protocol was achieved through redocking of the original ligands, yielding root mean square deviation (RMSD) values of less than 2.0 Å, confirming the reliability of the docking parameters. The best binding poses of genistein and kaempferol were analyzed for interaction patterns and visualized using PyMOL2 [60]. These simulations provided insight into the binding affinities and molecular interactions of the active compounds with their respective enzyme targets.

### 3.10. Statistical Analysis

All assays were performed in triplicate, and the results are presented as mean ± standard deviation (SD). Statistical analysis was conducted using one-way analysis of variance (ANOVA), followed by Tukey’s honestly significant difference (HSD) test for multiple comparisons. All analyses were carried out using SPSS software (version 17.0), with statistical significance set at *p* < 0.05.

## 4. Conclusions

The findings revealed that although the leaf extract of *C. juncea* exhibited the highest extraction yield and total flavonoid content, the root extract contained the highest total phenolic content and demonstrated superior antioxidant activity. Importantly, the root extract showed pronounced anti-aging potential by effectively inhibiting collagenase, elastase, hyaluronidase, and tyrosinase, outperforming both the leaf and flower extracts. LC-MS/QTOF analysis identified key antioxidant-related flavonoids—particularly genistein and kaempferol—in the root extract. These results were further supported by molecular docking simulations, which confirmed strong binding affinities between these compounds and enzymes associated with skin aging. Moreover, cytotoxicity and anti-inflammatory assays demonstrated the root extract’s non-toxic profile and its ability to suppress inflammatory mediators. Collectively, these findings underscore the potential of *C. juncea* root extract as a multifunctional natural bioactive agent for use in cosmetic and cosmeceutical formulations. For further study, we intend to quantify key antioxidant-related flavonoids, particularly genistein and kaempferol, evaluate the stability of the extract, develop an appropriate dosage form for topical application, and conduct clinical trials in human volunteers to support its potential use in cosmetic and dermatological products.

## Figures and Tables

**Figure 1 ijms-26-07716-f001:**
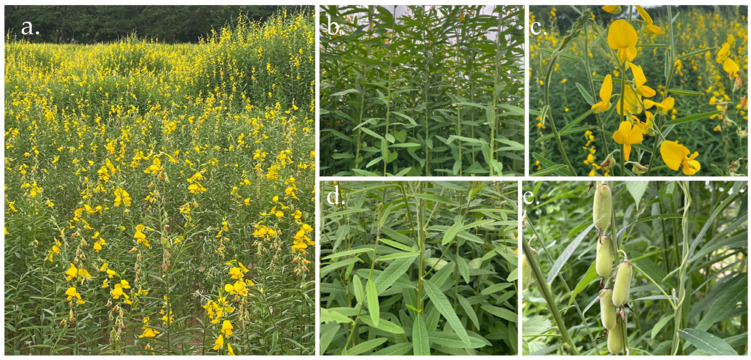
Botanical features of Sunn hemp: habit (**a**), stem and leaf arrangement (**b**), flowers (**c**), leaves (**d**), and fruits (**e**).

**Figure 2 ijms-26-07716-f002:**
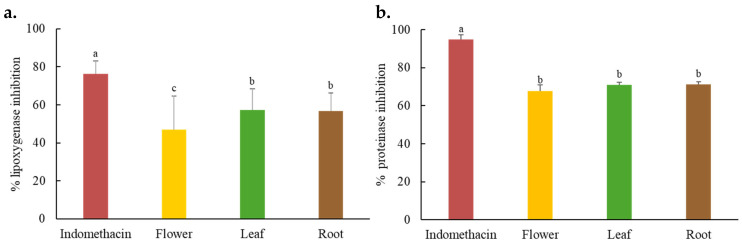
Lipoxygenase inhibitory (**a**) and proteinase inhibitory (**b**) activities of *C. juncea* extracts (concentration of 1 mg/mL). Data are presented as mean ± S.D. (*n* = 3). Different superscript letters (a–c) indicate a statistically significant difference (*p* < 0.05) among samples using one-way ANOVA, followed by Tukey’s multiple comparison post-hoc test.

**Figure 3 ijms-26-07716-f003:**
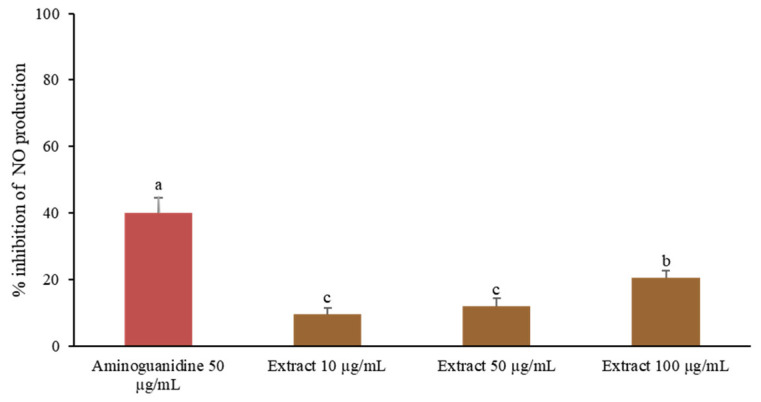
Inhibition of nitric oxide production was assessed in LPS-stimulated RAW 264.7 cells incubated with the *C. juncea* root extract at various concentrations. Different superscript letters (a–c) indicate a statistically significant difference (*p* < 0.05) among samples using one-way ANOVA, followed by Tukey’s multiple comparison post-hoc test.

**Figure 4 ijms-26-07716-f004:**
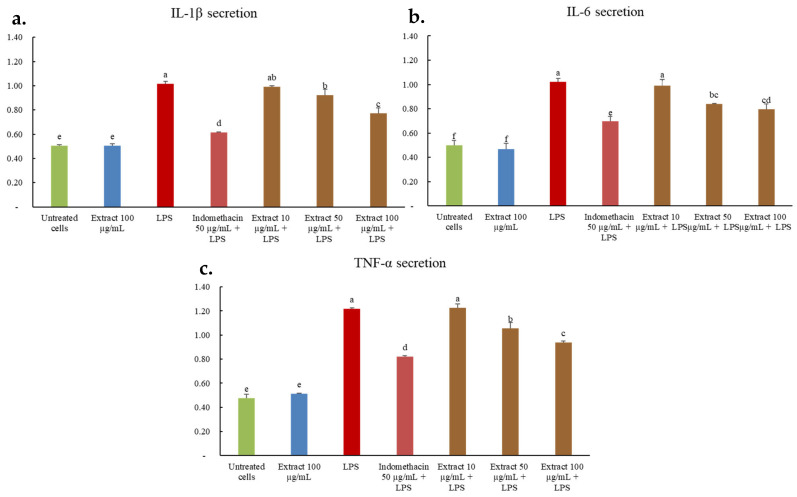
Mean percentage of effects of the root extract on mRNA expression of IL-1β (**a**), IL-6 (**b**), and TNF-α (**c**) from LPS-stimulated RAW264.7 cells, where β-actin is a house-keeping gene. Data are expressed as mean ± SD (*n* = 3), and superscript letters (a–f) indicate a statistically significant difference (*p* < 0.05) between groups using one-way ANOVA, followed by Tukey’s multiple comparison post-hoc test.

**Figure 5 ijms-26-07716-f005:**
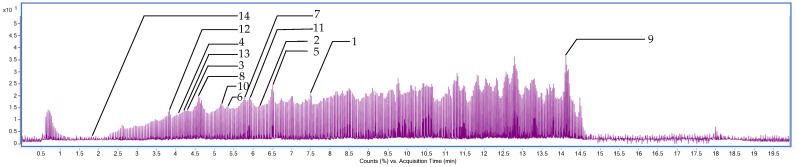
Chromatographic profile of *C. juncea* root extract obtained by liquid chromatography–mass spectrometry/quadrupole time-of-flight (LC-MS/QTOF).

**Figure 6 ijms-26-07716-f006:**
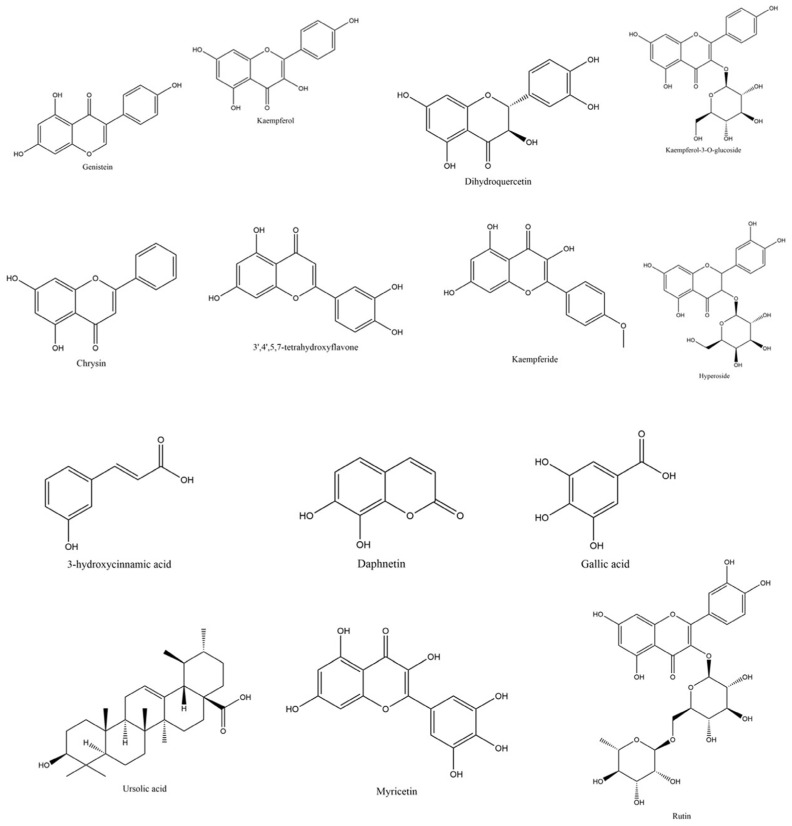
Chemical structures of flavonoids and flavonoid glycosides expected to be found in *C. juncea* root extract.

**Figure 7 ijms-26-07716-f007:**
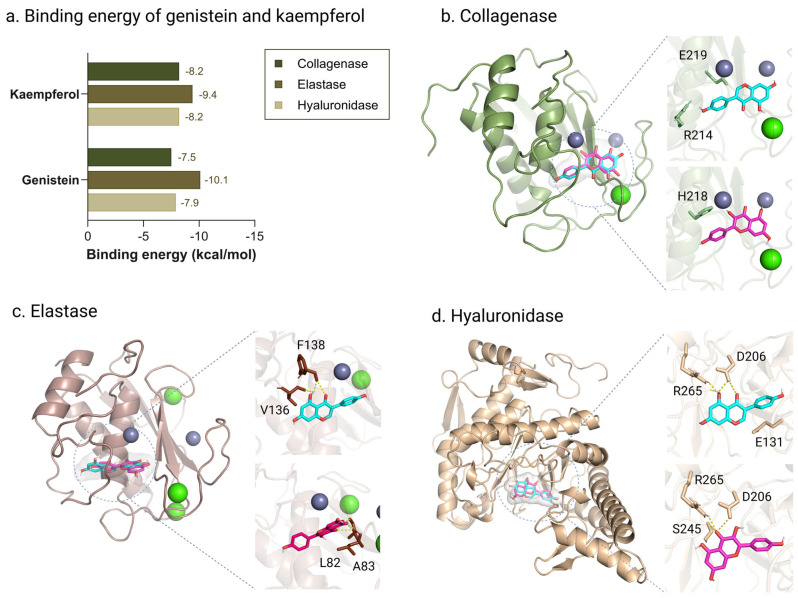
Binding scores and molecular interactions of genistein (cyan) and kaempferol (magenta) with collagenase, elastase, and hyaluronidase. Binding energies (kcal/mol) of genistein and kaempferol with collagenase, elastase, and hyaluronidase (**a**). Molecular interactions of genistein and kaempferol with key residues in collagenase (**b**), elastase (**c**), and hyaluronidase (**d**), showing hydrogen bond formation (yellow dash). Green spheres represent calcium ions, and grey spheres represent zinc ions.

**Table 1 ijms-26-07716-t001:** Antioxidant activity of *C. juncea* extracts evaluated using DPPH, FRAP, and β-carotene bleaching assays.

Samples	DPPH Radical Scavenging Activity (IC_50_ (mg/mL))	FRAP Value (mg FeSO_4_/g Extract)	β-Carotene Bleaching (% Inhibition)
Trolox	0.007 ± 0.00 ^a^	134.35 ± 1.18 ^a^	89.86 ± 6.72 ^a^
Flower extract	0.78 ± 0.01 ^d^	6.77 ± 0.10 ^c^	90.99 ± 1.27 ^a^
Leaf extract	0.64 ± 0.01 ^c^	7.17 ± 0.49 ^c^	89.19 ± 5.84 ^a^
Root extract	0.52 ± 0.06 ^b^	15.21 ± 1.46 ^b^	90.99 ± 7.09 ^a^

Different superscript letters (a–d) indicate significant differences in each assay by one-way ANOVA with multiple comparisons test using Tukey (*p* < 0.05).

**Table 2 ijms-26-07716-t002:** Anti-tyrosinase activity of *C. juncea* extracts (concentration of 2.5 mg/mL).

Samples	L-Tyrosine as Substrate (% Inhibition)	L-DOPA as Substrate (% Inhibition)
Kojic acid	98.47 ± 0.24 ^a^	98.58 ± 0.15 ^a^
Flower extract	26.62 ± 2.03 ^bc^	18.94 ± 2.84 ^c^
Leaf extract	15.45 ± 12.15 ^c^	43.91 ± 9.17 ^bc^
Root extract	33.56 ± 4.21 ^b^	44.15 ± 4.03 ^b^

Different superscript letters (a–c) indicate significant differences in each assay by one-way ANOVA with multiple comparisons test using Tukey (*p* < 0.05).

**Table 3 ijms-26-07716-t003:** Anti-aging activity of *C. juncea* extracts (concentration of 0.5 mg/mL).

Samples	Collagenase Inhibitory Activity (% Inhibition)	Elastase Inhibitory Activity (% Inhibition)	Hyaluronidase Inhibitory Activity (% Inhibition)
Ascorbic acid	85.44 ± 4.12 ^a^	86.51 ± 11.70 ^a^	ND
Tannic acid	ND	ND	97.59 ± 2.89 ^a^
Flower extract	15.28 ± 0.78 ^b^	30.34 ± 2.67 ^c^	1.72 ± 4.91 ^b^
Leaf extract	19.95 ± 10.10 ^b^	65.04 ± 13.61 ^b^	4.48 ± 3.16 ^b^
Root extract	91.60 ± 0.66 ^a^	67.17 ± 9.34 ^b^	8.50 ± 2.53 ^b^

ND: not determined. Different superscript letters (a–c) indicate a statistically significant difference (*p* < 0.05) in each assay using one-way ANOVA, followed by Tukey’s multiple comparison post-hoc test.

**Table 4 ijms-26-07716-t004:** Phytochemicals identified in *C. juncea* root extract by LC-MS/QTOF data in the negative ion mode.

No	Retention Time (min)	*m*/*z*	Tentative Identification	Formula	Mass Error (ppm)	Ion Species
1	7.569	269.04825	Genistein	C_15_H_10_O_5_	2.38	[M−H]^−^
2	6.299	285.04547	Kaempferol	C_15_H_10_O_6_	−16.07	[M−H]^−^
3	4.249	303.05389	Dihydroquercetin	C_15_H_12_O_7_	−9.57	[M−H]^−^
4	4.102	447.09470	Kaempferol-3-O-glucoside	C_21_H_20_O_11_	−2.89	[M−H]^−^
5	6.540	253.05211	Chrysin	C_15_H_10_O_4_	−5.85	[M−H]^−^
6	5.307	285.04224	3′,4′,5,7-tetrahydroxyflavone	C_15_H_10_O_6_	−6.21	[M−H]^−^
7	5.823	299.05591	Kaempferide	C_16_H_12_O_6_	8.66	[M−H]^−^
8	4.671	463.08795	Hyperoside	C_21_H_20_O_12_	0.54	[M−H]^−^
9	14.277	455.35516	Ursolic acid	C_30_H_48_O_3_	−6.50	[M−H]^−^
10	5.171	177.01897	Daphnetin	C_9_H_6_O_4_	−0.90	[M−H]^−^
11	5.902	317.02997	Myricetin	C_15_H_10_O_8_	1.04	[M−H]^−^
12	3.617	163.03993	3-hydroxycinnamic acid	C_9_H_8_O_3_	0.86	[M−H]^−^
13	4.208	609.14624	Rutin	C_27_H_30_O_16_	−0.20	[M−H]^−^
14	1.866	169.01414	Gallic acid	C_7_H_6_O_5_	−0.77	[M−H]^−^

## Data Availability

All the data generated or analyzed during this study are included in this published article.

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
