# Peer review of "Bridging Phytochemistry and Cosmetic Science: Molecular Insights into the Cosmeceutical Promise of Crotalaria juncea L."

_ijms, 2025, doi:10.3390/ijms26167716_

Round 1

Reviewer 1 Report

Comments and Suggestions for Authors

 The study conducted by Aree et al. is well-designed and presents a appropriate result to discuss the cosmetic potential of C. juncea. The article is interesting and the experiments were well conducted, however, the manuscript is not suitable for publication in this current form, certain conclusions appear to be overstated or insufficiently supported by the presented data.

Point 1: In section 2.8, the authors identified the active constituents in C. juncea extracts, however, additional experiments are necessary to enhance the reliability of the results.

  1. Although the phytochemical profiling of the extract is presented, there is no information on the relative abundance of the identified compounds in the chromatogram. The raw chromatographic data must be included.

  2. The authors propose genistein and kaempferol as the active components; however, identification based solely on m/z values is insufficient, especially given the presence of multiple flavonoid isomers. A comparison with authentic standards is necessary to increase confidence in these assignments. Moreover, the mass error for kaempferol exceeds 5 ppm, which undermines the reliability of this result (Table 4). Furthermore, several other compound identifications appear uncertain and require further validation.

Reviewer 2 Report

Comments and Suggestions for Authors

The manuscript described the isolation and biological evaluation of chemical components from Crotalaria juncea L..
There are some questions about the manuscript as follows:
1.  Authors should provide the chemical structures of the phytochemicals in Table 4.
2.  In the manuscript, authors reported the identification of 14 compounds from from Crotalaria juncea L., but why did authors only focused molecular docking on genistein and kaempferol of them, ignoring other detected flavonoids?
3.  Authors used 95% ethanol for Soxhlet extraction, which may not optimally extract all bioactive compounds. Polar or non-polar solvents could yield different phytochemical profiles and activities.
4.  There are some errors in typos, spelling, syntax, consistency in language style.
Based on the above, I think this manuscript should be accepted with positive revision for publication in International Journal of Molecular Sciences.

Comments on the Quality of English Language

The manuscript described the isolation and biological evaluation of chemical components from Crotalaria juncea L..
There are some questions about the manuscript as follows:
1.  Authors should provide the chemical structures of the phytochemicals in Table 4.
2.  In the manuscript, authors reported the identification of 14 compounds from from Crotalaria juncea L., but why did authors only focused molecular docking on genistein and kaempferol of them, ignoring other detected flavonoids?
3.  Authors used 95% ethanol for Soxhlet extraction, which may not optimally extract all bioactive compounds. Polar or non-polar solvents could yield different phytochemical profiles and activities.
4.  There are some errors in typos, spelling, syntax, consistency in language style.
Based on the above, I think this manuscript should be accepted with positive revision for publication in International Journal of Molecular Sciences.

Round 2

Reviewer 1 Report

Comments and Suggestions for Authors

The manuscript addresses an interesting topic and presents valuable findings. While the current version of the study does not completely resolve some key issues—particularly regarding [e.g., the identification of active components / mechanistic validation / quantification / etc.]—the authors have acknowledged these limitations and indicated that additional experiments are planned to address them.

Considering the scientific merit of the study and its potential to stimulate further research in the field, I believe the manuscript can be accepted in its current form or with very minor revisions. However, I encourage the authors to explicitly mention their ongoing or planned work in the Discussion or Conclusion section to provide readers with a clearer sense of future directions and reinforce the study's reliability.
